| Open Peer Review | Epidemiology | Methods and Protocols

# Diagnosis of fasciolosis antibodies in Brazilian cattle through ELISA employing both native and recombinant antigens

Guilherme Drescher,[1] Hellen Geremias dos Santos,[2] Mariane Marques da Guarda Pinto,[1] Luis Gustavo Morello,[3,4] Fabiano Borges Figueiredo[1]

**ABSTRACT** Bovine fasciolosis is a parasitic disease with a global reach. Coprological based on egg detection in fecal samples and liver inspection to evaluate the presence of the parasite is currently the gold standard for diagnosing chronic fasciolosis in cattle. However, these techniques are labor-intensive and ineffective during the acute phase of the disease. Serodiagnosis using native and recombinant antigens has become an interesting alternative in efforts to identify cattle fasciolosis. We evaluated cattle from abattoir ($n = 139$) and farms ($n = 500$) through liver inspection and coprological examination, respectively. Our laboratory team optimized and validated enzyme-linked immunosorbent assay tests based on somatic antigen, excretory/secretory proteins, and the recombinant antigen cathepsin L-1 to detect serum antibodies against fasciolosis in cattle. For animals from abattoir, 10 were positive for fasciolosis according to liver inspection. Both *Fh*ES and *Fhr*CL-1 presented an area under the receiver operating characteristic (AUROC) curve of 0.80, with a sensitivity of 0.80 (95% CI: 0.46–0.95) and 0.70 (95% CI: 0.38–0.90) and specificity of 0.81 (95% CI: 0.73–0.87) and 0.87 (95% CI: 0.80–0.92), respectively. For those cattle from farms, 28 were positive only for fasciolosis according to coprological examination. In this scenario, *Fh*ES gave the best performance, with an AUROC of 0.84, sensitivity of 0.79 (95% CI: 0.60–0.90), and specificity of 0.86 (95% CI: 0.82–0.89). In conclusion, our study highlights the potential of serodiagnosis for accurately screening cattle fasciolosis. The promising sensitivity and specificity values of *Fh*ES when compared to liver inspection or coprological examination enhance its importance for cattle fasciolosis diagnosis.

**IMPORTANCE** The aim of this article was to identify antibodies against fasciolosis in cattle in Brazil. The methodology was reproduced in our laboratory and applied for the first time to the Brazilian cattle herd. The antigens tested can be used as a screening test and thus speed up the diagnosis of bovine fascioliasis.

**KEYWORDS** fasciolosis, cattle, native antigens (*Fh*ES and *Fh*SA), recombinant antigen (*Fhr*CL-1), ELISA

*F*asciola hepatica, a plant-borne trematode species, is responsible for the zoonotic disease known as fasciolosis or liver fluke disease in humans and animals (1–3). The disease has traditionally been characterized as important in the veterinary context due to the substantial production and economic losses it causes in livestock (4–7). Herbivorous mammal hosts, such as cattle, goats, and sheep, are the most important disease transmission path to humans (8). Human fasciolosis is considered a neglected tropical disease by the World Health Organization, with estimates of 2.4 million infected individuals and 180 million people at risk of infection worldwide (9, 10).

This trematode has an extensive global distribution and is found on every continent except Antarctica. Human fasciolosis poses major health problems in Europe, Cuba,

Address correspondence to Guilherme Drescher, guidrescher@yahoo.com.br, or Fabiano Borges Figueiredo, fabiano.figueiredo@fiocruz.br.

The authors declare no conflict of interest.

Oceania, and the Americas (1, 11), with a higher number of cases reported in South America (Bolivia, Peru, Chile, Ecuador, and Venezuela) than in other regions (3, 12–14). In contrast, non-Andean, lowland countries in South America have reported sporadic and isolated human cases, including Uruguay (15) and Brazil (16, 17). Among animals, studies in the Americas have demonstrated a wide prevalence in goats and a lower prevalence in cattle (18, 19). In the Brazilian state of Santa Catarina, a prevalence of 10.8% in cattle was documented in an abattoir (20). Fasciolosis causes economic losses related to cattle production and severely impacts public health (6, 20, 21). Such economic losses have been quantified at a national level in Brazil, with a 5.8% reduction in carcass weight translating to a 35 USD loss per animal in this country (22, 23).

*F. hepatica* is adaptable to different environmental conditions and has the ability to switch hosts (24), resulting in a broad host range (10). Its spread is also related to the geographic expansion of its original intermediate host, the Lymnaeidae snails (1). The life cycle of this disease comprises three stages, each characterized by distinct symptoms. The acute phase, initiated through ingestion of metacercariae in contaminated vegetation and water, lasts 2–4 months and manifests as abdominal pain, fever, urticaria, and gastrointestinal disturbances (2, 25). The latent phase involves newly encysted juveniles penetrating the intestinal wall and peritoneum, migrating to the liver tissue, and reaching the bile ducts over several months (26–28). In the chronic phase, mature parasites in bile ducts produce eggs, causing severe liver and bile duct damage.

The established diagnostic method for bovine fasciolosis is the identification of eggs in feces (coprological examination), which is cost-effective and the gold standard for various parasitic diseases in humans and animals (27, 29). Diagnosis throughout coprological examination often occurs during the chronic phase, when much of the liver damage has already occurred (28). However, there is a consensus that this method is not completely reliable for several reasons. A period of 8–15 weeks post-infection is required for *F. hepatica* eggs to appear in feces when many pathological lesions have already manifested (30, 31). Additionally, the method may not detect low-intensity or intermittent infections (27, 32). In regions where the disease is not endemic, infections with immature flukes are not detected. Furthermore, the eggs are released intermittently from the bile ducts, so stool samples from infected patients (humans and animals) may not contain eggs (27).

Postmortem worm counting in the liver can be considered a valuable diagnostic method if the livers are appropriately sliced and soaked. *F. hepatica* can also be identified by inspecting cattle livers for adult worms in abattoirs. Postmortem examination of the bovine liver is a key approach to assessing the severity of *F. hepatica* infections. This entails examining livers for juvenile worms and bile ducts for adults, along with any associated pathological changes. Different rates of bovine fasciolosis have been reported in different abattoirs globally, with Brazil, for instance, documenting a 29.51% infection rate among animals (33). However, even mild or prepatent infections can evade detection, impacting the estimated sensitivity and specificity of the test.

Serological techniques, including lateral flow assays (34) and the indirect enzyme-linked immunosorbent assay (ELISA) (35–38), have been explored for detecting specific antibodies. ELISA-based detection of serum antibodies is a widely used diagnostic tool. It is highly regarded for its sensitivity and reliability in diagnosing acute infections, and it can complement fecal analysis for diagnosing latent and chronic infections (27). The antigens traditionally employed in serological tests consist of native antigens (somatic antigens and excretory/secretory antigens) of *F. hepatica* (35). To enhance diagnostic specificity, several purified *F. hepatica* antigens and recombinant antigens (36, 37) have been used, most notably cathepsin L, a major protease involved in bovine fasciolosis. Serological tests have demonstrated high accuracy in diagnosing human, bovine, and ovine fasciolosis. The recombinant cathepsin L1 test utilizes recombinant pro-cathepsin L1 and targets antibodies against cathepsin, a cysteine protease, for diagnosing fasciolosis caused by *F. hepatica* (37, 39), with no reported cross-reactions. Similarly, other studies observed no cross-reactions in native antigens and cathepsin-based ELISA

tests, reporting strong performance (39–44). While many serological methods have been published, only a few have been commercially adopted.

In this context, the present study aimed to assess the potential of available native antigens, both somatic (*Fh*SA) and excretory/secretory (*Fh*ES), and the recombinant antigen cathepsin L (*Fhr*CL-1) for serodiagnosis of cattle fasciolosis in Brazil.

## MATERIALS AND METHODS

### Characteristics of the cattle included in the study

#### Abattoir cattle sample

A total of 139 serum samples were collected from a cattle abattoir located in southern Santa Catarina, Brazil. The presence of cattle fasciolosis was determined through liver inspection. According to this approach, 10/139 (7.2%) cattle were diagnosed with *F. hepatica*, with no other parasites investigated during the veterinary inspection. Serum samples were processed, divided into aliquots, and stored at −30°C for subsequent ELISA testing.

#### Farms cattle sample

Five hundred serum and fecal samples (420 from female and 80 from male cattle) were obtained from 37 farms in southern Santa Catarina, Brazil. The samples were collected from cattle ranging from 6 months to 20 years old. Fecal samples (6 g) were used for *in vivo* diagnostics of fasciolosis and other parasites through coprological examination based on a sedimentation protocol (32). The tests were conducted in triplicate, and the entire sediments were analyzed under a stereomicroscope (32, 45). Serum samples were processed, divided into aliquots, and stored at −30°C for subsequent ELISA testing.

#### FhSA and FhES

The *Fh*SA and *Fh*ES preparations were carried out as follows: intact and live adult parasites were obtained from cattle livers at a local abattoir. Initially, the parasites underwent a series of 3–4 washes at room temperature using 0.01 M phosphate-buffered saline (PBS) with a pH of 7.2 to eliminate any traces of blood and bile.

For the *Fh*SA preparation, the parasites were kept in a PBS solution and transported to the laboratory. Subsequently, the parasites were macerated and divided into separate portions. The protease inhibitor trans-Epoxysuccinyl-L-leucylamido(4-guanidino) butane (E-64; Sigma-Aldrich, US) was added to each sample at a concentration of 10 µM to minimize protein degradation. The antibiotics penicillin (100 U/mL) and streptomycin (0.25 mg/mL) were also incorporated to counteract bacterial activity.

For the *Fh*ES preparation, parasites were incubated in Roswell Park Memorial Institute (RPMI) 1640 medium at 37°C for 6 h. Within the laboratory setting, the parasites were subjected to five washing rounds with PBS containing antibiotics (penicillin and streptomycin). The first two washes used a volume of 10 mL PBS with antibiotics, while the subsequent three used a volume of 8 mL. Subsequently, the parasites were transferred using forceps into a 15 mL falcon tube containing RPMI 1640 medium preheated to 37°C. They were then cultured at a concentration of six parasites per 3 mL for 6 h at 37°C.

After incubation, the falcon tube was centrifuged at 14,000 × *g* for 30 min. The supernatant was then collected and divided into three microtubes, each containing 1 mL. E-64 was introduced to prevent protein degradation. The secretory/excretory antigens were obtained by culturing *F. hepatica* in RPMI medium and filtered using an Amicon Ultra-15 100 kDa centrifugal filter (Millipore, UK). During the antigen filtration process from the excretory/secretory systems, the RPMI medium was replaced with a saline buffer.

SDS-PAGE was conducted to analyze the protein content within *Fh*SA and *Fh*ES. Quantification of both the somatic antigen and the excretory/secretory antigens was

carried out using a fluorimetric method in a Qubit (Thermo Fischer Scientific, US) instrument. Following protein quantification, the supernatants of *Fh*SA and *Fh*ES were divided into aliquots and stored at −30°C until use.

## Expression and purification of FhrCL-1

The full-length cDNA of *F. hepatica* preprocathepsin L1 (U62288.2) was obtained commercially in the pPIC9K vector from (GenScript, US). Protein expression was conducted using the multicopy system of the *Pichia pastoris* GS115 strain. The recombinant sequence featured a single amino acid substitution, replacing the active site Cys25 with Gly. This alteration resulted in the loss of functional activity while preserving the enzyme's conformation, rendering it more stable during fermentation and downstream isolation processes (39, 46, 47).

To generate the inactive enzyme, fermentation was performed in a liquid minimal medium containing yeast extract and glycerol (BMGY) to enhance yeast cell density. Cultivation in BMGY took place for 16 h at 30°C with agitation at 250 rpm. Once the yeast cell density reached an OD600 of 2–6, approximately 1 mL of the inoculum was transferred to a liquid minimal medium containing yeast extract and methanol (BMMY) to induce *Fh*rCL1 expression. Cultivation in BMMY lasted 92 h at 30°C under agitation at 250 rpm. During this time, the medium was supplemented with 1% methanol every 24 h.

After completing the cultivation period, the culture was centrifuged at 10,000 × $g$ for 30 min at room temperature. The resulting pellets were discarded. *Fh*rCL-1 was isolated from the supernatant using Ni-NTA affinity chromatography, following previously described methods (39, 47, 48).

## ELISA optimization and development

### FhSA, FhES, and FhrCL-1

To define ELISA conditions, we performed a matrix comparison using various antigen concentrations, dilutions of the primary sera, and dilutions of secondary antibodies for *Fh*SA, *Fh*ES, and *Fh*rCL-1 antigens, respectively.

Optimal antigen concentrations and serum dilutions were determined by checkerboard titrations. *Fh*SA, *Fh*ES, and *Fh*rCL-1 antigens (0.5 µg/mL, 1.0 µg/mL, and 1.0 µg/mL, respectively) were dissolved separately in bicarbonate/carbonate coating buffer at pH 9.0 and added to each ELISA plate (Sarstedt AG & Co. KG, DE). One hundred microliters of the solution were then added to each well and incubated overnight at 4°C. The plates were washed three times with 0.05% Tween-80 in water. After coating, an additional blocking step with 100 µL 1% skimmed milk in 0.05% Tween-80 was performed for 1 h at 37°C. After a further washing procedure, 100 µL of sera-diluted pooled samples were added to each antigen (1:50, 1:100, and 1:50, respectively), and the plates were incubated for 1 h at 37°C. Following another wash, 100 µL of peroxidase-conjugated anti-bovine antibody (Sigma-Aldrich, US) for each antigen (1:10.000, 1:10.000, and 1:30.000, respectively) was added to the wells, and the plates were incubated for 30 min at 37°C. After a final washing step, bound antibodies were detected by adding 100 µL of tetramethylbenzidine (Thermo Fischer Scientific, US). After incubation at room temperature in the dark for 10–20 min, the reaction was stopped with 50 µL of 0.1 M sulfuric acid. The plates were read on an ELISA reader at 450 nm to determine absorbance values.

After developing and optimizing serological ELISA conditions, we tested serum samples from cattle collected in an abattoir and cattle farms. Negative and positive controls were used to diagnose fasciolosis in cattle by ELISA, using *Fh*SA, *Fh*ES, and *Fh*rCL-1 as antigens. A pool of four samples (two negative samples for the presence of fasciolosis in the visceral inspection and two negative samples for the coprological examination) was used as a negative control on each plate. As a positive control, a pool of four samples was used on each plate (two positive samples for the presence of fasciolosis in the visceral inspection and two positive samples for the coprological examination). Positive control, negative control, and plate control were used in duplicate in all experiments.

## Statistical analysis

To evaluate the diagnostic performance of native (*Fh*SA and *Fh*ES) and recombinant antigens (*Fh*rCL-1), we used liver inspection and coprological examination as the gold standard test for cattle from abattoir and farms, respectively. Initially, the distribution of the quantitative values for the serodiagnosis tests was analyzed according to the categories (positive or negative) of the gold standard tests, aiming to explore their descriptive statistics, such as minimum, maximum, and median values, first and third quartile, mean values and SD, as well as to inspect for outliers.

The optimal cutoff value for each ELISA method was based on a logistic regression model, considering as response variable the gold standard test results (positive or negative) and as predictor the log of the quantitative values for the serodiagnosis test. Briefly, we applied a logistic regression model to adjust a classifier and a leave-one-out cross-validation (CV) technique to evaluate its diagnostic performance in data not used for its adjustment. Thus, on each CV iteration, the observations were divided into training and test data; the former was used to adjust a logistic model and the latter to estimate the probability of being classified as a positive sample. After all samples were part of the training and test data, the vector of estimated probabilities was used to evaluate the diagnostic performance of the model. For this, it was necessary to choose a cutoff point for the estimated probability, aiming to classify samples as positive or negative. We chose the cutoff that maximizes the model's sensitivity and specificity and calculated the area under the receiver operating characteristic (AUROC) curve, sensitivity (S), specificity (E), positive predictive values (PPVs), and negative predictive values (NPVs) and the respective 95% CI of all of these estimates. The analyses were performed separately for cattle from abattoir and farms on R software using caret, pROC, and CompareTests packages. The script used for the analysis is available at https://github.com/Hellengeremias/Fasciolosis.

## RESULTS

### Abattoir cattle sample

Table 1 shows a summary of the *Fh*ES, *Fh*SA, and *Fh*rCL-1 values according to the presence (positive group) or absence (negative group) of fasciolosis detected by liver inspection of cattle from the abattoir ($n = 139$). In general, for the three tests, the positive group ($n = 10$) had higher values for the first and third quartiles as well for median and mean than the negative group ($n = 129$).

**TABLE 1** Descriptive summary of the three tests when applied to cattle from abattoirs ($n = 139$)[a]

| Summary values | Native antigens (OD) | | Recombinant antigen (OD) |
|---|---|---|---|
| | *Fh*ES | *Fh*SA | *Fh*rCL-1 |
| Positive group ($n = 10$) | | | |
| Minimum | 0.360 | 0.196 | 0.076 |
| First quartile | 0.490 | 0.270 | 0.111 |
| Median | 0.571 | 0.443 | 0.252 |
| Mean (SD) | 0.573 (0.141) | 0.439 (0.186) | 0.235 (0.126) |
| Second quartile | 0.668 | 0.568 | 0.336 |
| Maximum | 0.815 | 0.716 | 0.436 |
| Negative group ($n = 129$) | | | |
| Minimum | 0.192 | 0.114 | 0.057 |
| First quartile | 0.298 | 0.169 | 0.078 |
| Median | 0.362 | 0.222 | 0.090 |
| Mean (SD) | 0.393 (0.133) | 0.263 (0.144) | 0.104 (0.064) |
| Second quartile | 0.452 | 0.303 | 0.107 |
| Maximum | 0.806 | 1.247 | 0.638 |

[a] SD= Standard Deviation; OD= Optical Density.

The AUROC for *Fh*ES, *Fh*SA, and *Fhr*CL-1 adjusted models was 0.80 (95% CI: 0.67–0.92), 0.74 (95% CI: 0.55–0.93), and 0.80 (95% CI: 0.61–0.98), respectively (Fig. 1). For each test, we chose the cutoff point that maximizes the model's sensitivity and specificity (Table 2). The three tests had a moderately accurate performance. The chosen cutoff value for the *Fh*ES ELISA test showed higher sensitivity and NPV (0.80 and 0.98, respectively), indicating the test suitability in screening fasciolosis: only 2 of the 10 fasciolosis cases were mistakenly classified as negative (false negative result). Thus, out of 106 negative results, 104 were true negative. Nonetheless, since the fasciolosis prevalence is low (10 positive cases in 139 cattle), we observed a large number of false positives and consequently a low PPV: only 8 of the 33 positive results were true positive, suggesting the serological tests cannot be used to confirm the presence of the disease.

## Farms cattle sample

The coprological examination resulted in 405/500 (81%) negative and 95/500 (19%) positive results. Of the 95 positive results, 28/500 (5.6%) were positive only for *F. hepatica* eggs, and 10/500 (2%) for *F. hepatica* and other parasites: 7/500 (1.4%) also contained *Strongylidae* eggs, 2/500 (0.4%) *Eimeria* eggs, and 1/500 (0.2%) *Strongylidae* and *Eimeria* eggs. The examination also showed that 44/500 (8.8%) cattle were positive only for *Strongylidae* eggs and 13/500 (2.6%) for both *Strongylidae* and *Eimeria* eggs. Animals positive for other parasites than *F. hepatica* (*n* = 67) were excluded from the diagnostic performance evaluation of native and recombinant antigens described below.

Table 3 shows the summary values of *Fh*ES, *Fh*SA, and *Fhr*CL-1 according to the presence (positive group) or absence (negative group) of fasciolosis detected by

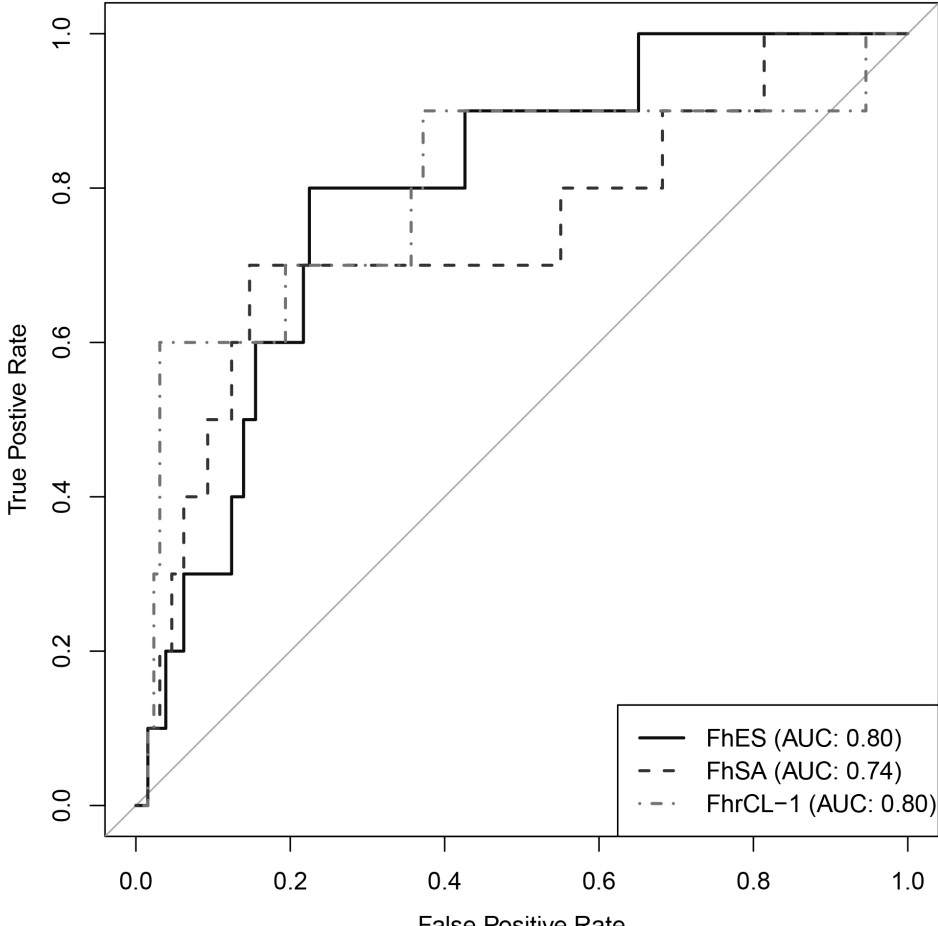

**FIG 1** Receiver operating characteristic curves for *Fh*ES, *Fh*SA, and *Fhr*CL-1 for cattle from abattoir (*n* = 139).

**TABLE 2** Diagnostic performance measures for the three tests on abattoir cattle considering liver inspection as the gold standard method (*n* = 139, 10 positive cases)[a]

| Performance measures | Native antigens | | Recombinant antigen |
|---|---|---|---|
| | *Fh*ES estimate (95% CI) | *Fh*SA estimate (95% CI) | *Fhr*CL-1 estimate (95% CI) |
| Cutoff | 0.4895 (OD) | 0.379 (OD) | 0.1050 (OD) |
| Sensitivity | 8/10 (0.80) | 7/10 (0.70) | 7/10 (0.70) |
| | (0.46–0.95) | (0.38–0.90) | (0.38–0.90) |
| Specificity | 104/129 (0.81) | 113/129 (0.88) | 112/129 (0.87) |
| | (0.73–0.87) | (0.81–0.92) | (0.80–0.92) |
| PPV | 8/33 (0.24) | 7/23 (0.30) | 7/24 (0.29) |
| | (0.17–0.34) | (0.19–0.45) | (0.18–0.43) |
| NPV | 104/106 (0.98) | 113/116 (0.97) | 112/115 (0.97) |
| | (0.94–0.99) | (0.94–0.99) | (0.94–0.99) |

[a]CI= Confidence Intervals; PPV= Positive Predictive Values; NPV= Negative Predictive Values; OD= Optical Density.

coprological examination in cattle from farms. The positive group (*n* = 28) had higher values for the first and third quartiles as well as for the median and mean for the three serological tests than the negative group (*n* = 405).

The AUROCs for *Fh*ES, *Fh*SA, and *Fhr*CL-1 adjusted models were 0.84 (95% CI: 0.76–0.93), 0.73 (95% CI: 0.61–0.85), and 0.67 (95% CI: 0.54–0.80), respectively (Fig. 2). For each test, we chose the cutoff point that maximizes the model's sensitivity and specificity (Table 4). For this scenario, the *Fh*ES also presented better results, with a sensitivity of 0.79 and an NPV of 0.98. Of the 353 negative results, 347 were true negatives when using the chosen cutoff value for the *Fh*ES-adjusted model. Nonetheless, since the disease prevalence was low (28/433, 6,5%), the cutoff value for the *Fh*ES adjusted model resulted in a higher number of false positives: out of the 80 positive results, only 22 were true positives.

Aiming to investigate the impact of cross-reactions on the diagnostic performance of the three antigens (*Fh*ES, *Fh*SA, and *Fhr*CL-1), we reanalyzed the data from farms also considering the 67 samples positive by coprological examination that were excluded for the analyzes presented previously. The 10 samples positive for fasciolosis and other parasites were considered in the positive group, together with the 28 samples positive only for fasciolosis, and the 57 samples positive only for other parasites were considered in the negative group, together with the 405 samples negative for fasciolosis. First,

**TABLE 3** Descriptive summary for the three tests when applied to cattle from farms (*n* = 433). Cattle diagnosed with other than parasites than *F. hepatica* (*n* = 67) were excluded from the analyses[a]

| Summary values | Native antigens (OD) | | Recombinant antigen (OD) |
|---|---|---|---|
| | *Fh*ES | *Fh*SA | *Fhr*CL-1 |
| Positive group (*n* = 28) | | | |
| Minimum | 0.213 | 0.168 | 0.058 |
| First quartile | 0.412 | 0.440 | 0.102 |
| Median | 0.529 | 0.637 | 0.141 |
| Mean (SD) | 0.560 (0.213) | 0.641 (0.299) | 0.184 (0.116) |
| Second quartile | 0.776 | 0.927 | 0.248 |
| Maximum | 0.828 | 1.312 | 0.454 |
| Negative group (*n* = 405) | | | |
| Minimum | 0.085 | 0.078 | 0.054 |
| First quartile | 0.202 | 0.266 | 0.088 |
| Median | 0.259 | 0.382 | 0.103 |
| Mean (SD) | 0.289 (0.122) | 0.406 (0.191) | 0.113 (0.044) |
| Second quartile | 0.353 | 0.510 | 0.126 |
| Maximum | 0.898 | 1.373 | 0.410 |

[a] SD= Standard Deviation; OD= Optical Density.

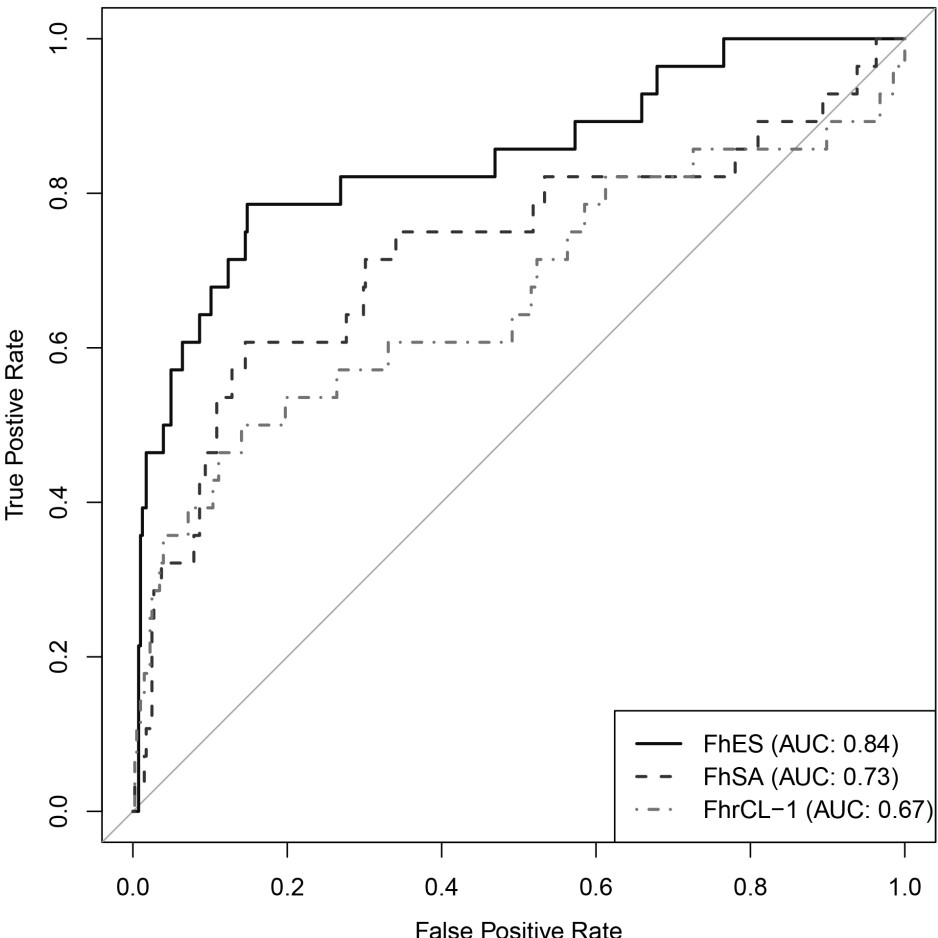

**FIG 2** Receiver operating characteristic curves for *Fh*ES, *Fh*SA, and *Fhr*CL-1 for cattle from farms (*n* = 433).

we evaluated the distribution of absorbance values of these 67 samples and those 28 positive only for fasciolosis and compared this distribution with the cutoff points obtained from the adjusted models that considered samples positive only for fasciolosis (Fig. S1). Overall, these results show that samples positive for fasciolosis presented higher absorbance values. Table S1 shows the summary values of *Fh*ES, *Fh*SA, and *Fhr*CL-1 according to the presence (positive group) or absence (negative group) of fasciolosis

**TABLE 4** Diagnostic performance measures for the three tests on farms cattle considering coprological examination as the gold standard method (*n* = 433, 28 positive cases). Cattle diagnosed with other than parasites than *F. hepatica* (*n* = 67) were excluded from the analyses[a]

| Performance measures | Native antigens | | Recombinant antigen |
|---|---|---|---|
| | *Fh*ES estimate (95% CI) | *Fh*SA estimate (95% CI) | *Fhr*CL-1 estimate (95% CI) |
| Cutoff | 0.4105 (OD) | 0.4830 (OD) | 0.1270 (OD) |
| Sensitivity | 22/28 (0.79) | 20/28 (0.71) | 16/28 (0.57) |
| | (0.60–0.90) | (0.52–0.85) | (0.39–0.74) |
| Specificity | 347/405 (0.86) | 285/405 (0.70) | 304/405 (0.75) |
| | (0.82–0.89) | (0.66–0.75) | (0.71–0.79) |
| PPV | 22/80 (0.28) | 20/140 (0.14) | 16/117 (0.14) |
| | (0.22–0.34) | (0.11–0.18) | (0.10–0.19) |
| NPV | 347/353 (0.98) | 285/293 (0.97) | 304/316 (0.96) |
| | (0.97–0.99) | (0.95–0.98) | (0.94–0.97) |

[a]CI= Confidence Intervals; PPV= Positive Predictive Values; NPV= Negative Predictive Values; OD= Optical Density.

diagnosed by coprological examination, and Table S2 shows the results of the diagnostic performance of the adjusted models. The AUROC curve for the *Fh*ES, *Fh*SA, and *Fhr*CL-1 was similar to those that considered samples positive only for fasciolosis: 0.83 (95% CI: 0.75–0.90), 0.73 (95% CI: 0.63–0.83), and 0.68 (95% CI: 0.58–0.79), respectively (Fig. S2).

## DISCUSSION

Our study is the first to compare native and recombinant antigens for diagnosing cattle fasciolosis in Brazilian animals. Coprological and liver inspection were used as the gold standard diagnostic tests for both farms and abattoir animals. The *Fh*ES serological test was better able to discriminate positive and negative samples for both farms and abattoir animals, and it seems suitable for screening purposes. The economic and public health problems caused by cattle fasciolosis have been reported in different parts of the world, including Brazil (20, 21, 28, 49). It is important to develop and establish a reliable, simple, and rapid diagnostic tool for properly diagnosing cattle fasciolosis in Brazil, especially in endemic areas. In the present study, we evaluated the performance of three ELISA tests using *Fh*ES, *Fh*SA, and *Fhr*CL-1 antigens to diagnose cattle fasciolosis based on information obtained from a meta-analysis study (50).

Liver necropsy, which diagnoses fasciolosis when bile ducts are dissected, is the only conclusive diagnostic procedure for *F. hepatica* (7, 51). This is impractical as a herd or flock management tool, as it can only be carried out postmortem. Specific ELISA tests for liver flukes have been developed to meet these requirements and are now routinely used for cattle (30, 52). ELISAs for *F. hepatica* are versatile tests capable of detecting specific antibodies or antigens in fecal samples as well as pooled or individual milk and sera (36, 37, 53). One significant drawback of relying on fecal egg counts is the inability to diagnose immature migrating stages of liver flukes within the final host. Consequently, using ELISA tests with early diagnostic potential represents a notable advantage (27, 36). The most detrimental phase of this infection occurs during the migration of immature stages (36, 37). The application of ELISA techniques for *F. hepatica* diagnosis has consistently exhibited enhanced sensitivity compared to coprological methods (36–38). Moreover, it offers the distinct advantage of detecting pre-patent infections.

Serological diagnosis of cattle fasciolosis based on fractions of adult worm antigens has been reported in different studies worldwide (36, 43, 54). To this end, we used two cattle populations with known infection status (the presence of eggs in the feces or parasites in the liver). Our first serological panel comprised more than 100 cattle samples collected in an abattoir. A small number of articles that evaluated the diagnosis of fasciolosis in cattle used samples collected in abattoirs (36). Our second serological panel consisted of 500 samples of blood and feces from cattle collected on farms. The studies that evaluated the serological diagnosis of bovine fasciolosis used small panels with up to 100 animals (43, 54–56).

A critical point for evaluating a new immunodiagnostic test is to propose a cutoff point that properly discriminates between negative and positive samples. The absorbance values of *Fh*ES, *Fh*SA, and *Fhr*CL-1 antigens tested had a good ability to distinguish between positive and negative samples in abattoir samples. Only the *Fh*ES antigen performed well in differentiating positive and negative cattle fasciolosis on serum samples collected on farms. Our investigation demonstrated that the absorbance values for the *Fh*ES antigen were comparable to those reported in other studies when sera from cattle with fasciolosis were examined using coprological testing as the gold standard (55).

Our study established a cutoff value for each proposed ELISA test based on positive and negative samples using liver inspection and coprological examination as the gold standard tests. The cutoff points for *Fh*ES, *Fh*SA, and *Fhr*CL-1 were 0.4895, 0.379, and 0.1050, respectively, for cattle from the abattoir and 0.4105, 0.4830, and 0.1270 for those from farms. The native antigens *Fh*ES and *Fh*SA consist of a complex mixture of proteins, potentially leading to elevated absorbance values. In contrast, the recombinant antigen *Fhr*CL-1 is a single purified protein, which could account for the comparatively lower

absorbance values observed. Different approaches are employed when developing ELISA tests for serological diagnosis of fasciolosis in cattle. The cutoff values reported by studies assessing one of these antigens vary, although they are often higher than those found in our analysis. Different methods based on the average absorbance value and the ROC curve are used in the ELISA tests created using native and recombinant antigens for the serological diagnosis of bovine fasciolosis (36–38, 55, 57).

Serology offers the advantage of earlier detection of infections in comparison to fecal egg detection. In addition, when compared to coprological methods, serological approaches, particularly the ELISA test, are very sensitive and specific. Since *F. hepatica* is the main cause of cattle fasciolosis, most of the studies related to the disease diagnosis focus on purified subunits from either *Fh*SA or *Fh*ES (native antigens) of this parasite species (36, 43, 58, 59). The cattle in this study come from farms in southern Santa Catarina, where the prevalence of the disease is considered low (20). Despite the observed low prevalence of the disease, the antigen *Fh*ES showed good diagnostic performance for both samples collected in the abattoir and farms, with sensitivities of 80% and 79% and specificities of 81% and 86%, respectively. Other studies that also used native antigens reported sensitivity ranging from 80% to 100% and specificity from 50% to 100% for serological diagnosis of bovine fasciolosis (37, 60).

Serological diagnosis for cattle fasciolosis using recombinant antigens (cathepsin and saposin) has been developed in the last years. Cathepsin is an important enzyme the parasite uses to elicit a humoral response in cattle as early as 2 weeks after infection (36, 38). In our study, the antigen *Fh*rCL-1 presented diagnostic performance as good as those observed in *Fh*ES for abattoir cattle.

Sera samples from farm cattle infected with other parasites were used to evaluate the impact of cross-reactivity in our ELISA tests. Cross-reactivity analysis is fundamental since fasciolosis is a worldwide parasitic disease that can co-occur with other cattle parasitic diseases. Furthermore, current parasitological methods depend on the worker's expertise because *F. hepatica* eggs can be confused with eggs from other helminths. Therefore, a good diagnostic test needs to be able to distinguish between *Fasciola* and other parasitic diseases. We did not observe substantial differences between the adjusted models without and with positive samples for other parasites, which suggests that the test differentiated animals positive for fasciolosis from cattle samples with other parasites.

In our study, the cattle in the positive group had positive fecal egg counts or the presence of *F. hepatica* in the liver, indicating that each animal was currently infected. Diagnosis of this infection is usually based on coprological techniques. The intermittent nature of the eggs' evacuation through the feces was the reason for the low sensitivity of the coproscopy in detecting fasciolosis in cattle (31). Moreover, a prolonged pre-patent period of 8–15 weeks after the infection is required for the eggs to be shed in the feces (27, 31). Compared to fecal egg counts, serology can detect infections 7–8 weeks earlier (36, 37) and is considered a very sensitive method (61), but it does not distinguish between current and past infections. Results indicated that indirect ELISA using *Fh*ES and *Fh*rCL-1 antigens could be an efficient and rapid diagnostic method for cattle fasciolosis compared to coprology. Therefore, using both methods together provided excellent information about the real infection situation. Of the three antigens (*Fh*SA, *Fh*ES, and *Fh*rCL-1) tested for the serological diagnosis of *F. hepatica* in cattle, the *Fh*ES presented satisfactory results in both scenarios, when compared to liver inspection in cattle from abattoir and to coprological examination in those from farms, suggesting it may be used for the development of ELISA tests for fasciolosis screening.

## Conclusion

We have developed three ELISAs utilizing two native antigens and one recombinant antigen for detecting *F. hepatica* antibodies. We validated these ELISAs using cattle serum samples collected from abattoir and farms, considering the liver inspection and coprological examination as gold standard tests, respectively. The ELISA test using *Fh*ES

as an antigen had good diagnostic performance in the two scenarios (abattoir and farms) for screening fasciolosis. Notably, the results were promising even in the face of the relatively low prevalence of cattle fasciolosis. The proposed ELISA test has the potential to be used in situations where it is more challenging to do a coprological investigation or examine the liver of cattle. These assays constitute a vital component of the immunodiagnostic toolkit that our laboratory is developing to improve the serodiagnosis of fasciolosis in Brazilian cattle. Recognizing that positive outcomes in antibody detection tests may not necessarily indicate ongoing infections but a history of exposure, we are actively exploring alternatives, such as an antigen detection ELISA using monoclonal antibodies. As a prospect, it is important to apply the test to more positive samples and also to explore cross-infection. Furthermore, ongoing research efforts are focused on adapting our in-house ELISA methods into more streamlined and dependable formats, such as immunochromatography or dot ELISA. This adaptation aims to facilitate potential commercialization and validation within Brazilian regions where the disease is endemic.

## ACKNOWLEDGMENTS

The authors would like to thank the Fundação Oswaldo Cruz (FIOCRUZ-PR), Coordenação de Aperfeiçoamento de Pessoal de Nível Superior (CAPES), and Conselho Nacional de Pesquisa (CNPq) for the possibility of developing this research. We also thank the FIOCRUZ Network of Technological Platforms for providing access to the Integrative Structural Biology facility at the Carlos Chagas Institute, FIOCRUZ-PR. We thank Wagner Nagib for providing the figure design and Jaqueline de Oliveira Rosa for the plasmid design. The authors would like to thank the Centro Universitário Barriga Verde (UNIBAVE).

This study was supported by Fundação Oswaldo Cruz (FIOCRUZ).

G.D., L.G.M., and F.B.F.: Study conception and design. G.D.: Conceptualization, methodology, manuscript writing, and drafting the manuscript. H.G. dos S.: Data curation, statistical analysis, and manuscript reviewing. M.M. da G.P.: Manuscript reviewing. L.G.M.: Manuscript reviewing. F.B.F.: Manuscript writing, reviewing, and editing. L.G.M. and F.B.F. supervised the study. All authors contributed to the article and approved the submitted version.

## AUTHOR AFFILIATIONS

[1]Cellular Biology Laboratory, Carlos Chagas Institute, Oswaldo Cruz Foundation (FIOCRUZ-PR), Curitiba, Paraná, Brazil
[2]Carlos Chagas Institute, Oswaldo Cruz Foundation (FIOCRUZ-PR), Curitiba, Paraná, Brazil
[3]Laboratory for Applied Science and Technology in Health, Carlos Chagas Institute, Oswaldo Cruz Foundation (FIOCRUZ-PR), Curitiba, Paraná, Brazil
[4]Parana Institute of Molecular Biology, Curitiba, Paraná, Brazil

## AUTHOR ORCIDs

Guilherme Drescher  http://orcid.org/0000-0001-6331-3862
Fabiano Borges Figueiredo  http://orcid.org/0000-0001-6861-0997

## AUTHOR CONTRIBUTIONS

Guilherme Drescher, Conceptualization, Data curation, Methodology, Writing – original draft, Writing – review and editing | Hellen Geremias dos Santos, Data curation, formal analysis, Writing – review and editing | Mariane Marques da Guarda Pinto, Methodology, Writing – original draft | Luis Gustavo Morello, Supervision, Writing – original draft, Writing – review and editing | Fabiano Borges Figueiredo, Conceptualization, Data curation, Funding acquisition, Project administration, Supervision, Validation, Visualization, Writing – original draft, Writing – review and editing

## ETHICS APPROVAL

The protocols and methods used were approved by the Ethics Committee of the Centro Universitário Barriga Verde (UNIBAVE; protocol CAAE: 82403017.4.0000.5598).

## ADDITIONAL FILES

The following material is available online.

### Supplemental Material

**Figure S1 (Spectrum00095-24-S0001.tiff).** Distribution of absorbance values.
**Figure S2 (Spectrum00095-24-S0002.tiff).** ROC curves.
**Tables S1 and S2 (Spectrum00095-24-S0003.docx).** Diagnostic performance measures for the three tests on farms cattle ($n = 500$, 38 positive cases).

### Open Peer Review

**PEER REVIEW HISTORY (review-history.pdf).** An accounting of the reviewer comments and feedback.

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
