## [Reviewer comments · Microbiology Spectrum]

Microbiology Spectrum

Diagnosis of fasciolosis antibodies in Brazilian cattle through enzyme-linked immunosorbent assay (ELISA) employing both native and recombinant antigens

GUILHERME DRESCHER, Hellen dos Santos, Mariane Pinto, Luis Morello, and Fabiano Figueiredo

Corresponding Author(s): GUILHERME DRESCHER, Instituto Carlos Chagas

Review Timeline:

Submission Date:	January 9, 2024
Editorial Decision:	February 11, 2024
Revision Received:	March 8, 2024
Accepted:	March 11, 2024

Editor: Artem Rogovskyy

Reviewer(s): Disclosure of reviewer identity is with reference to reviewer comments included in decision letter(s). The following individuals involved in review of your submission have agreed to reveal their identity: Kobra Mokhtarian (Reviewer #1)

Transaction Report:

DOI: <https://doi.org/10.1128/spectrum.00095-24>

Re: Spectrum00095-24 (Diagnosis of anti-fasciolosis antibodies in Brazilian cattle through enzyme-linked immunosorbent assay (ELISA) employing both native and recombinant antigens)

Dear Dr. GUILHERME DRESCHER:

Thank you for the privilege of reviewing your work. Below you will find my comments, instructions from the Spectrum editorial office, and the reviewer comments.

Revision Guidelines

Sincerely,
Artem Rogovskyy
Editor
Microbiology Spectrum

Reviewer #1 (Comments for the Author):

The following paragraph should be rewritten. The results show whether it is in the stool or in the serum. Please rewritten clearly. It is difficult to distinguish the eggs of Fasciola species by microscopic observation. How was the diagnosis made?

Reviewer #2 (Comments for the Author):

- The number of sample size is confusing, there are 500, 433 and 139. The number of samples should be the same or tallying.
2. The Number of positives should indicated and in brackets, you put the positive percentage eg 111/139 (80%) unlike only 80% or 0.80. This can be misleading to some readers.
 3. The Confidence interval should be recalculated, it is misleading or misreading. This is a range of 95% CI 80% (72-85%) OR 95% CI 0.80 (0.72-0.85) OR 95% CI 0.80 (0.72; 0.85)

**Diagnosis of anti-fasciolosis antibodies in Brazilian cattle through enzyme-linked**
**immunosorbent assay (ELISA) employing both native and recombinant antigens**

Guilherme Drescher^{a*}, Hellen Geremias dos Santos^b, Mariane Marques da Guarda
Pinto^a, Luis Gustavo Morello^{c,d}, Fabiano Borges Figueiredo^a

6 ^a Cellular Biology Laboratory, Carlos Chagas Institute, Oswaldo Cruz Foundation
(FIOCRUZ-PR), Curitiba 81310-020, Brazil.

8 ^b Carlos Chagas Institute, Oswaldo Cruz Foundation (FIOCRUZ-PR), Curitiba 81310-020,
Brazil.

10 ^c Laboratory for Applied Science and Technology in Health, Carlos Chagas Institute,
Oswaldo Cruz Foundation (FIOCRUZ-PR), Curitiba, 81310-020, Brazil.

12 ^d Parana Institute of Molecular Biology, Curitiba, 81310-020, Brazil.

***Correspondence author:** Guilherme Drescher (guidrescher@yahoo.com.br) and
Fabiano Borges Figueiredo (fabiano.figueiredo@fiocruz.br).

Abstract

Bovine fasciolosis is a parasitic disease with global reach. Coprological based on egg
detection in fecal samples and liver inspection to evaluate the presence of the parasite is
currently the gold standard for diagnosing **cattle fasciolosis**. However, these techniques
are labor-intensive and ineffective during the acute phase of the disease. Serodiagnosis
using native and recombinant antigens has become an interesting alternative in efforts to
identify cattle fasciolosis. In the present study, we evaluated cattle from abattoirs through
liver inspection and from farms through coprological examination. Our laboratory team
optimized and validated an ELISA to detect serum anti-fasciolosis antibodies in cattle.
This assay utilized native antigens, somatic antigen and excretory/secretory proteins, in
addition to the recombinant antigen cathepsin L-1. The native antigens were derived from
parasite, the recombinant antigen was produced in the laboratory. For animals from
abattoirs, both *FhES* and *FhrCL-1* presented an area under the ROC curve of 0.80, with
a sensitivity of 0.80, (95% CI 0.46; 0.95) and 0.70 (95% CI 0.38; 0.90) and specificity of
0.81 (95% CI 0.73; 0.87) and 0.87 (95% CI 0.80; 0.92), respectively. *FhES* gave the best
performance for those originating from the farm, with an AUROC of 0.84, sensitivity of
0.79 (95%CI 0.60; 0.90), and specificity of 0.86 (95%CI 0.82; 0.89). In conclusion, our
study highlights the potential of serodiagnosis for accurately screening cattle fasciolosis.
When comparing liver inspection and coprological examination, the promising sensitivity
and specificity values of *FhES* highlight its importance as a tool for cattle fasciolosis
diagnosis.

Keywords: Fasciolosis, cattle, native antigens (*FhES* and *FhSA*), recombinant antigen
(*FhrCL-1*), ELISA.

*Introduction*

[revised manuscript text omitted]

In this context, the present study aimed to assess the potential of available native
antigens, both somatic (*FhSA*) and excretory/secretory (*FhSE*), and the recombinant
antigen cathepsin L (*FhrCL-1*) for serodiagnosis of **cattle fasciolosis in Brazil.**

*Materials and methods*

The protocols and methods used were approved by the Ethics Committee of the
Evandro Chagas Institute (INI/FIOCRUZ) (protocol CAAE: 10324719.6.0000.5262).

*Characteristics of the cattle included in the study*

Five hundred serum and fecal samples (420 from females and 80 from males)
were obtained from 37 cattle farms in southern Santa Catarina. The samples were
collected from cattle ranging in age from six months to 20 years. Fecal samples (6g) were
used for in vivo diagnostics of cattle fasciolosis through coprological examination based
on a sedimentation protocol (32). The tests were conducted in triplicate, and the entire
sediments were analyzed under a stereomicroscope (32,45).

**We found 405 negative and 95 positive results for eggs in the fecal samples. Of**
**the 95 positive animals, 38 (7.6%) were positive for *F. hepatica*, 28 for *F. hepatica* eggs**
**only, and 10 for *F. hepatica* and other parasites: seven also contained eggs of strongylid**

[revised manuscript text omitted]

*Results*

**Table 1 shows** a summary of the *FhES*, *FhSA*, and *FhrCL-1* values according to
the presence (positive group) or absence (negative group) of fascioliasis detected by liver
inspection of cattle from the abattoir (n=139). In general, for the three tests, the positive
group had higher values for the first and third quartiles as well for median and mean than
the negative group.

*Insert Table 1*

The AUROC for *FhES*, *FhSA*, and *FhrCL-1* adjusted models were 0.80 (95%CI:
0.67; 0.92), 0.74 (95%CI: 0.55; 0.93) and 0.80 (95%CI: 0.61; 0.98), respectively (Figure
1). For each test, we chose the cutoff point that maximizes the model's sensitivity and

specificity (Table 2). The three tests had a moderately accurate performance, although
*FhES* showed higher sensitivity and NPV (0.80 and 0.98, respectively), indicating that the
test is suitable for fasciolosis screening, as it had a low frequency of false negative results:
two of approximately 10 cases of the disease were mistakenly classified as negative.
Thus, out of 106 negative results, only two were false. Even though the specificity of the
tests was high, the fact that the disease occurs rarely (7.2%) means that the tests cannot
be used to confirm the presence of the disease due to the large number of false-positive
results (for example, for *FhES*, 25 of 33 positive results were false).

***Insert Table 2 and Figure 1***

Table 3 shows the summary values of *FhES*, *FhSA*, and *FhrCL-1* according to the
presence (positive group) or absence (negative group) of fasciolosis detected by
coprological examination in cattle from farms. The positive group had higher values for
the first and third quartiles as well as for the median and mean for the three serological
tests than the negative group.

***Insert Table 3***

The AUROCs for *FhES*, *FhSA*, and *FhrCL-1* adjusted models were 0.84 (95%CI:
0.76; 0.93), 0.73 (95%CI: 0.61; 0.85), and 0.67 (95%CI: 0.54; 0.80), respectively (Figure
2). For each test, we chose the cutoff point that maximizes the model's sensitivity and
specificity (Table 4). For this scenario, the *FhES* also presented better results, with a
sensitivity of 0.79 and an NPV of 0.98. Of the 353 negative results, only six were false
negatives when using the chosen cutoff value for the *FhES*. This confirms its suitability
for screening fasciolosis. Despite the high specificity, the disease prevalence was only
5.6%, out of the 80 positive results, and 58 were false positive results.

***Insert Table 4 and Figure 2***

Aiming to investigate the impact of cross-reactions on the diagnostic performance
of the three antigens (*FhES*, *FhSA*, and *FhrCL-1*), for the 95 positive samples (according
to the coprological examination), we evaluated the distribution of absorbance values and
the cut-off points obtained from the adjusted models that considered samples positive for
fascioliasis only (Supplementary Figure 1). Overall, these results show that positive
samples for fasciolosis had higher absorbance values. We also adjusted the logistic
regression models by considering cattle samples from farms positive for parasitic
infections other than fasciolosis (n=500). Supplementary Table 1 shows the summary
values of *FhES*, *FhSA*, and *FhrCL-1* according to the presence (positive group) or
absence (negative group) of fasciolosis and the Supplementary Table 2, the results of the
adjusted models. The AUROC curve for the *FhES*, *FhSA*, and *FhrCL-1* were similar to
those that considered only samples positive for fascioliasis: 0.83 (95%CI: 0.75; 0.90),
0.73 (95%CI: 0.63; 0.83), and 0.68 (95%CI: 0.58; 0.79), respectively (Supplementary
Figure 2).

[revised manuscript text omitted]

*Conflict of Interest*

The authors declare that the research was conducted in the absence of any commercial or
financial relationships that could be construed as a potential conflict of interest.

*Author Contributions*

GD, LGM, and FBF: Study conception and design. GD: Conceptualization, methodology,
manuscript writing, and drafting the manuscript. HG dos S: Data curation, meta-analysis, and
manuscript reviewing. MM da GP: Manuscript reviewing. LGM: Manuscript reviewing. FBF:
Manuscript writing, reviewing, and editing. LGM and FBF supervised the study. All authors
contributed to the article and approved the submitted version.

*Funding*

This study was supported by Fundação Oswaldo Cruz (FIOCRUZ).

*Acknowledgments*

The authors would like to thank the Fundação Osvaldo Cruz (FIOCRUZ-PR), Coordenação de
Aperfeiçoamento de Pessoal de Nível Superior (CAPES), and Conselho Nacional de Pesquisa
(CNPq) for the possibility of developing this research. We also thank the FIOCRUZ Network of
Technological Platforms for providing access to the Integrative Structural Biology facility at the
Carlos Chagas Institute, FIOCRUZ-PR. We thank Wagner Nagib for providing the figure design
and Jaqueline de Oliveira Rosa for the plasmid design. The authors would like to thank the Centro
Universitário Barriga Verde (UNIBAVE).

*References*

[revised manuscript text omitted]

	Native antigens		Recombinant antigen
Summary values	FhES	FhSA	FhrCL-1
Positive group (n= 10)			
Minimum	0.360	0.196	0.076
1st quartile	0.490	0.270	0.111
Median	0.571	0.443	0.252
Mean (SD)	0.573 (0.141)	0.439 (0.186)	0.235 (0.126)
2nd quartile	0.668	0.568	0.336
Maximum	0.815	0.716	0.436
Negative group (n= 129)			
Minimum	0.192	0.114	0.057
1st quartile	0.298	0.169	0.078
Median	0.362	0.222	0.090
Mean (SD)	0.393 (0.133)	0.263 (0.144)	0.104 (0.064)
2nd quartile	0.452	0.303	0.107
Maximum	0.806	1.247	0.638

Legend: SD= Standard Deviation.

Table 2. Diagnostic performance measures for the three tests by considering the presence of the parasite in the liver as the gold standard method (n=139).

	Native antigens		Recombinant antigen
Performance measures	FhES estimate (95%CI)	FhSA estimate (95%CI)	FhrCL-1 estimate (95%CI)
Cutoff	0.4895	0.379	0.1050
Sensitivity	0.80 (0.46; 0.95)	0.70 (0.38; 0.90)	0.70 (0.38; 0.90)
Specificity	0.81 (0.73; 0.87)	0.86 (0.81; 0.92)	0.87 (0.80; 0.92)
PPV	0.24 (0.17; 0.34)	0.30 (0.19; 0.45)	0.29 (0.18; 0.43)
NPV	0.98 (0.94; 0.99)	0.97 (0.94; 0.99)	0.97 (0.94; 0.99)

Legend: CI= confidence intervals; PPV= positive predictive values; NPV= negative predictive values.

Table 3. Descriptive summary for the three tests when applied to cattle from farms (n=433).

	Native antigens		Recombinant antigen
Summary values	FhES	FhSA	FhrCL-1
Positive group (n= 28)			
Minimum	0.213	0.168	0.058
1st quartile	0.412	0.440	0.102
Median	0.529	0.637	0.141
Mean (SD)	0.560 (0.213)	0.641 (0.299)	0.184 (0.116)
2nd quartile	0.776	0.927	0.248
Maximum	0.828	1.312	0.454
Negative group (n= 405)			
Minimum	0.085	0.078	0.054
1st quartile	0.202	0.266	0.088
Median	0.259	0.382	0.103
Mean (SD)	0.289 (0.122)	0.406 (0.191)	0.113 (0.044)
2nd quartile	0.353	0.510	0.126
Maximum	0.898	1.373	0.410

Legend: SD= Standard Deviation.

Table 4. Diagnostic performance measures for the three tests by considering the coprological examination as the gold standard method (n=433).

	Native antigens		Recombinant antigen
Performance measures	FhES estimate (95%CI)	FhSA estimate (95%CI)	FhrCL-1 estimate (95%CI)
Cutoff	0.4105	0.4830	0.1270
Sensitivity	0.79 (0.60; 0.90)	0.71 (0.52; 0.85)	0.57 (0.39; 0.74)
Specificity	0.86 (0.82; 0.89)	0.70 (0.66; 0.75)	0.75 (0.71; 0.79)
PPV	0.28 (0.22; 0.34)	0.14 (0.11; 0.18)	0.14 (0.10; 0.19)
NPV	0.98 (0.97; 0.99)	0.97 (0.95; 0.98)	0.96 (0.94; 0.97)

Legend: CI= confidence intervals; PPV= positive predictive values; NPV= negative predictive values.

Figure 1. ROC curves for FhES, FhSA and FhrCL-1 for cattle from abattoirs (n=139)

Figure 2. ROC curves for FhES, FhSA and FhrCL-1 for cattle from farms (n=433)

Reviewer's comments

	Line number	Current	Suggestion or Comments	Reasons
1	1	Diagnosis of anti-fasciolosis antibodies in Brazilian cattle through enzyme-linked immunosorbent assay (ELISA) employing both native and recombinant antigens	Diagnosis of fasciolosis in Brazilian cattle through enzyme-linked immunosorbent assay (ELISA) employing both native and recombinant antigens”	The title should change because we are detecting the disease using antibodies through ELISA (serological technique), hence we cannot say diagnosis of anti-fasciolosis antibodies.
2	24-25	In the present study, we evaluated cattle from abattoirs through liver inspection and from farms through coprological examination.	In the present study, we evaluated cattle from abattoirs and farms through liver inspection and coprological examination respectively	Repetition of wording
3	Abstract	ROC curve of 0.80, with a sensitivity of 0.80, (95% CI 0.46; 0.95) and 0.70 (95% CI 0.38; 0.90)	ROC curve of 0.80, with a sensitivity of 0.80, (95% CI 0.46- 0.95) and 0.70 (95% CI 0.38- 0.90) OR ROC curve of 0.80, with a sensitivity of 0.80, (95% CI 0.46, 0.95) and 0.70 (95% CI 0.38, 0.90)	The CI should be written as a range. However, check this range it's too big from 46% to 95% the other one from 38% to 90%
	66translating to a 35 USD loss per head in this countrytranslating to a 35 USD loss per animal in this country	Per head may be confusing to other readers.
	85	In regions in which the disease is not endemic, infections.....	In regions, where the disease is not endemic, infections....	
	106	bovine fascioliasis	Bovine fasciolosis	Be consistent
	125	500 samples	How did the authors come up with this samples size? Any sample size formula?	
	127	collected from cattle ranging in age from six months to 20 years.	...collected from cattle ranging from six months to 20 years old	
	131-142	We found 405 negative and 95 positive results for eggs in the fecal samples. Of 132 the 95 positive animals, 38 (7.6%) were positive for F. hepatica , 28 for F. hepatica eggs only, and 10	Move to results section	Because this section is for methodology only and not results.

		for F. hepatica and other parasites: seven also contained eggs of strongylid genera, two Eimeria eggs, and one Strongylidae and Eimeria eggs. The examination also showed that 44 (8.8%) cattle were positive for strongylid genera only and 13 (2.6%) for both strongylid genera and Eimeria. Animals positive for other parasites than F. hepatica were excluded from the diagnostic performance evaluation of native and recombinant antigens. A total of 139 serum samples were collected from a cattle abattoir located in southern Santa Catarina. The presence of cattle fasciolosis was determined through liver inspection. According to this approach, 10 (7.2%) cattle were diagnosed with F. hepatica, with no other parasites investigated during the veterinary inspection.		
	131	38 were positive for F. hepatica	Were the 38 included in the 139?	
	Table 2	0.80 (0.46; 0.95), 0.70 (0.38; 0.90), 0.70 (0.38; 0.90)	Recalculate all the ranges of CI, For example 0.80 (0.72-0.85); 0.7 (0.62-0.77) Also include the positive figures unlike only percentage positivity 80% of 139 is 111/139 (80%)	They look to be too wide and appears to be out of range, considering (n=139)
	Table 4	0.79(0.60;0.90)	Recalculate all the ranges of CI, e.g (0.79 (0.75-0.82	Some of them look to be too wide and appears to be out of range, considering (n=433)
		The number of samples differ. You have 500 in initial stage but you are reporting 433	The samples collected should all be analysed. Where are other 67 samples?	
		Only 139 serum and 500 fecal samples	For serum samples you analysed only 139 why not	

			analysed serum from all 500 animals so that you compare the results for feacal examination and serology results?	
	143	Serum samples collected from both the abattoir and farms were processed, divided into aliquots, and stored at -30°C for subsequent ELISA testing.	This suggest that you had 500 serum samples why conduct ELISA on 139 only?	

Generally the research is okay but needs major revision. The number samples are confusing there 500 then 433 and 139. The number of samples should the same or tallying.

2. The Number of positive should indicated and in brackets you put the positive percentage eg 111/139 (80%) unlike only 80% or 0.80. This can be misleading to some readers.

3. The Confidence interval should be recalculated it is misleading or misreading. This is a range 95% CI 80% (72-85%) OR 95% CI 0.80 (0.72-0.85) OR 95% CI 0.80 (0.72; 0.85)

We thank the reviewer for their helpful suggestions that improved the message of our manuscript. Please find in below our point-to-point answers and commentaries on some text improvements we have made. In the revised version of our manuscript, major changes/insertions are outlined in red.

Independent Review Report, Reviewer 2

1. Line 1. **Current:** “Diagnosis of anti-fasciolosis antibodies in Brazilian cattle through enzyme-linked immunosorbent assay (ELISA) employing both native and recombinant antigens”/ **Suggestion:** “Diagnosis of fasciolosis in Brazilian cattle through enzyme-linked immunosorbent assay (ELISA) employing both native and recombinant antigens.”

RESPONSE: thank you for mentioning this. We modified the title as suggested by the Reviewer.

2. Lines 24–25. **Current:** “In the present study, we evaluated cattle from abattoirs through liver inspection and from farms through coprological examination”/ **Suggestion:** “In the present study, we evaluated cattle from abattoirs and farms through liver inspection and coprological examination respectively.”

RESPONSE: we appreciate this comment. The text was rewritten as suggested by the Reviewer.

3. Abstract. **Current:** “ROC curve of 0.80, with a sensitivity of 0.80 (95% CI 0.46; 0.95) and 0.70 (95% CI 0.38; 0.90)”/ **Suggestion:** “ROC curve of 0.80, with a sensitivity of 0.80, (95% CI 0.46- 0.95) and 0.70 (95% CI 0.38- 0.90).” **Comment:** The CI should be written as a range. However, check this range it’s too big from 46% to 95% the other one from 38% to 90%.

RESPONSE: thank you for mentioning this. We modified de CI written as suggested by the Reviewer. Regarding the reviewer's observation about the size of the confidence interval (very large), we reviewed the analyzes performed and confirmed the values obtained. For the confidence interval estimation, we user CompareTests() function from CompareTests R package. In the current version of the manuscript, we include a link to an open access repository where the script used in data analysis was made available. These results are for the analyzes of abattoir data that includes 139 animals, of which 10 were positive for fasciolosis according to liver inspection. As the sample is small and the prevalence of the disease is low, we expected a wider (less precise) confidence interval, especially for the sensitivity measure. To aid in the interpretation of the performance measures presented, in the current version of our manuscript we highlight the sample size and the number of positive cases in the abstract and in the results section for each group we studied.

4. Line 66. **Current:** “...translating to a 35 USD loss per head in this country”/ **Suggestion:** “...translating to a 35 USD loss per animal in this country.”

RESPONSE: thank you for pointing this out. The sentence was rewritten as suggested by the Reviewer.

5. Line 85. **Current:** “In regions in which the disease is not endemic, infections...”/
Suggestion: “In regions, where the disease is not endemic, infections....”

RESPONSE: thank you for pointing this out. The sentence was rewritten as suggested by the Reviewer.

6. Line 106. **Current:** “bovine fascioliasis”/ **Suggestion:** “bovine fasciolosis.”

RESPONSE: we appreciate this comment and apologize for the mistake. The sentence was rewritten according to the suggested correction.

7. Line 125. **Current:** “...500 samples”/ **Comment:** “How did the authors come up with this samples size? Any sample size formula?”

RESPONSE: thank you for mentioning this. Serum and fecal samples obtained from farms (n=500), as well as serum samples obtained from abattoir (n=139) represent convenience samples, and therefore, were not obtained from probabilistic sampling plans, as they included the voluntary participation of regional agricultural establishments in the State of Santa Catarina, located in the south of Brazil.

8. Line 127. **Current:** “...collected from cattle ranging in age from six months to 20 years”/**Suggestion:** “...collected from cattle ranging from six months to 20 years old.”

RESPONSE: thank you for pointing this out. The sentence was rewritten as suggested by the Reviewer.

9. Lines 131–142. **Current:** “We found 405 negative and 95 positive results for eggs in the fecal samples. Of 132 the 95 positive animals, 38 (7.6%) were positive for *F. hepatica*, 28 for *F. hepatica* eggs only, and 10 for *F. hepatica* and other parasites: seven also contained eggs of strongylid genera, two *Eimeria* eggs, and one *Strongylidae* and *Eimeria* eggs. The examination also showed that 44 (8.8%) cattle were positive for strongylid genera only and 13 (2.6%) for both strongylid genera and *Eimeria*. Animals positive for other parasites than *F. hepatica* were excluded from the diagnostic performance evaluation of native and recombinant antigens. A total of 139 serum samples were collected from a cattle abattoir located in southern Santa Catarina. The presence of cattle fasciolosis was determined through liver inspection. According to this approach, 10 (7.2%) cattle were diagnosed with *F. hepatica*, with no other parasites investigated during the veterinary inspection”/ **Comment:** “Move to results section. Because this section is for methodology only and not results.”

RESPONSE: we appreciate this comment. Now this paragraph has been moved to the results section as suggested by the Reviewer. Furthermore, we have made some improvements to the wording of this paragraph (lines 292-299) to clarify the two independent samples we analyzed (from the farms and the abattoir).

10. Line 131. **Current:** “38 were positive for *F. hepatica*”/ **Comment:** “Were the 38 included in the 139?”

RESPONSE: we appreciate this comment and apologize for the lack of clarity in the wording of the study design, which includes two cattle samples obtained independently,

one from farms and the other from abattoir. We rewrote the method section (lines 123-143), separating the description of the samples obtained in each context into subsections to make this information clearer. In this sense, the 38 positive samples for fasciolosis correspond to samples from farms (in which, in total, 500 animals were evaluated), where the disease was investigated through coprological examination (lines 136-143 in the methods section and lines 292-299 in the results section). Among the samples from the abattoir (which totaled 139 animals), 10 were positive for fasciolosis according to liver inspection (lines 127-132 in the methods section and lines 268-272 in the results section).

11. Line 143. **Current:** "Serum samples collected from both the abattoir and farms were processed, divided into aliquots, and stored at -30°C for subsequent ELISA testing"/
Comment: "This suggest that you had 500 serum samples why conduct ELISA on 139 only?"

RESPONSE: thank you for mentioning this. As described in the previous item, we apologize for the lack of clarity in the description of the samples used in this research. We reiterate that improvements were made in the writing of the methods section to overcome this issue (lines 123-143). ELISA tests were conducted on both groups: on the 500 samples obtained on the farms and on the 139 obtained at abattoir. In the results section, Tables 1 and 2, and Figure 1, show the analyzes of the ELISA data for cattle at abattoir, and Tables 3 and 4, and Figure 2 correspond to the analyzes of the ELISA data for cattle on farms. These results covered 433 animals, since those that presented positive results in the coprological examination for parasites other than *F. hepatica* were excluded (n=67). Nonetheless, supplementary material shows the analyzes of all farms data (i.e., including those animals that tested positive for other parasites in the coprological examination, totaling 500 cattle) aiming to evaluate the impact of cross-infection on the diagnostic performance of serodiagnosis tests.

12. Table 2. **Current:** "...0.80 (0.46; 0.95), 0.70 (0.38; 0.90), 0.70 (0.38; 0.90)"/
Suggestion: "Recalculate all the ranges of CI, For example 0.80 (0.72-0.85); 0.7 (0.62-0.77). Also include the positive figures unlike only percentage positivity 80% of 139 is 111/139 (80%). They look to be too wide and appears to be out of range, considering (n=139) bovine fascioliasis."

RESPONSE: we appreciate this comment and apologize for the lack of clarity in the description of the estimated diagnostic measures, presented on Table 2 for cattle from abattoir and on Table 4 and on Supplementary Table 2 for those from farms. We added on these Tables the numerator and denominator correspondent to the estimate of each diagnostic measure. For example, in Table 2, the estimated sensitivity of 0.80 for *FhES* is derived from the ratio of 8 true positive cases identified by the chosen cutoff point for *FhES* optical density, divided by the 10 positive cases identified by the gold standard method (liver inspection). Also, we checked all the estimated diagnostic measures we presented on Tables 2 and 4 and Supplementary Table 2, as well as their confidence interval estimation. As mentioned on item 3, the wider confidence interval for sensitivity measure is expected due to the rare disease occurrence on the studied samples. Also,

we include a link to an open access repository on the current version of the manuscript where the script used in the data analysis was made available.

13. Table 4. **Current:** "...0.79(0.60;0.90)" / **Suggestion:** "Recalculate all the ranges of CI, e.g (0.79 (0.75–0.82))."

RESPONSE: thank you for mentioning this. We apologize for the lack of clarity in the description of the estimated diagnostic measures, presented on Table 4 for cattle from farms. We correct the CI presentation as suggested by the Reviewer and added information regarding numerator and denominator of the estimated diagnostic measures. Please, see the answer presented in the previous item (item 12).

14. General comment. **Current:** "The number of samples differ. You have 500 in initial stage but you are reporting 433"/ **Comment:** "The samples collected should all be analyzed. Where are other 67 samples?"

RESPONSE: thank you for mentioning this. We apologize for the lack of clarity in the description of the samples used in this research. We reiterate that improvements were made in the writing of the methods section to overcome this issue (lines 123-143). Please, see the answer presented in the items 9, 10 and 11.

15. General comment. **Current:** "Only 139 serum and 500 feacal samples"/ **Comment:** "For serum samples you analysed only 139 why not analysed serum from all 500 animals so that you compare the results for feacal examination and serology results?"

RESPONSE: thank you for mentioning this. We apologize for the lack of clarity in the description of the samples used in this research. We reiterate that improvements were made in the writing of the methods section to overcome this issue (lines 123-143). Please, see the answer presented in the items 9, 10 and 11.

Generally, the research is okay but needs major revision.

1.The number samples are confusing there 500 then 433 and 139. The number of samples should the same or tallying.

RESPONSE: thank you for mentioning this. We apologize for the lack of clarity in the description of the samples used in this research. In the method section, the subsection entitled "Characteristics of the cattle included in the study" has been revised and restructured (lines 123-143) aiming to separately describe the two studied cattle groups, the one from farms and the other from abattoir.

2. The Number of positive should indicated and in brackets you put the positive percentage eg 111/139 (80%) unlike only 80% or 0.80. This can be misleading to some readers.

RESPONSE: thank you for mentioning this. We revised Tables 2 and 4 and Supplementary Table 2 aiming to add the numerator and denominator of the estimated diagnostic measures. Please, see the answer presented in the item 12.

3. The Confidence interval should be recalculated it is misleading or misreading. This is a range 95% CI 80% (72-85%) OR 95% CI 0.80 (0.72-0.85) OR 95% CI 0.80 (0.72; 0.85)

RESPONSE: thank you for mentioning this. We correct the CI presentation as suggested by the Reviewer. Also, we checked all the estimated diagnostic measures we presented on Tables 2, 4 and Supplementary Table 2, as well as their confidence interval estimation. Please, see the answer presented in the item 12.

We thank the reviewer for their helpful suggestions that improved the message of our manuscript. Please find in below our point-to-point answers and commentaries on some text improvements we have made. In the revised version of our manuscript, major changes/insertions are outlined in red.

Independent Review Report, Reviewer 1

1. Line 21. **Current:** “cattle fasciolosis”/ **Suggestion:** “chronic fasciolosis”.

RESPONSE: thank you for mention this. We replaced the term as suggested by the Reviewer.

2. *Fasciola gigantica* is more prevalent in Kettle, which species of *Fasciola* is used in this article? How is the *Fasciola* species confirmed?

RESPONSE: thank you for pointing this out. *Fasciola hepatica* is most prevalent in America. In our study, we employed both native and recombinant *F. hepatica* antigens.

For the native antigens, *F. hepatica* specimens was collected from a local cattle abattoir. The classification of these parasites as *F. hepatica* was based on morphological characteristics and parasite size during the veterinary inspection (lines 127-132).

In contrast, for the recombinant antigen, a genetic sequence encoding the cathepsin of *F. hepatica* retrieved from a genetic sequence bank was utilized (lines 181-187).

3. Lines 131–144. How did you differentiate the *Fasciola* species in Cattle? *Fasciola gigantica* is more common in cattle. How did you make a definitive diagnosis of *Fasciola hepatica*?

RESPONSE: thank you for bringing this to our attention. *F. hepatica* occurs more frequently in America. For the native antigens, the identification of *F. hepatica* was based on the morphological characteristics of the parasite. For the recombinant antigen, a genetic sequence coding for the *F. hepatica* antigen was used. The genetic sequence is described in the materials and methods of the manuscript (lines 181–187).

4. Line 223. In which temperature and condition? For TMB the time is 15-20 min. For your research, the color will be dim and therefore light absorption will be low.

RESPONSE: thank you for pointing this out. This sentence was rewritten (lines 220–223). After incubation at room temperature in the dark for 10- 20 min, the reaction was stopped with 50 μ L of 0.1 M sulfuric acid.

The sentence has been revised to reflect adjustments made in accordance with the ELISA protocol utilized. In light of TMB's photosensitivity, rigorous measures were instituted to ensure the accurate execution of the ELISA protocol. Detailed procedures have been outlined in the methods section of the manuscript.

Regarding the absorbance values highlighted by the reviewer, our hypothesis posits a plausible correlation between antigen composition and the observed variations in absorbance values. The heterogeneous protein composition of native antigens suggests a propensity for higher absorbance values. Conversely, the singular nature of the recombinant antigen may contribute to comparatively lower absorbance values.

5. Line 263. The purification of *FhSA*, *FhES*, and *FhrCL-1* proteins was shown on SDS-PAGE.

RESPONSE: thank you for bringing this to our attention. We conducted an analysis of the purification of both native and recombinant antigens using SDS-PAGE; data are not shown. We followed the purification methodology described in the methods section, lines 172-177 and 196-199 for the native and recombinant antigens, respectively.

6. Line 281. ... positive results were false.

RESPONSE: thank you for pointing this out. This sentence was rewritten (lines 282 - 285). Since the fasciolosis prevalence is low (10 positive cases in 139 cattle), we observed a large number of false positives and consequently a low PPV: only 8 of the 33 positive results were true positive, suggesting the serological tests cannot be used to confirm the presence of the disease.

7. Line 308. Overall, these results....

RESPONSE: thank you for pointing this out. This sentence was rewritten. Overall, these results show that samples positive for fasciolosis presented higher absorbance values (lines 329 - 330).

What sera were used for check the cross reaction? The sera of patients with hydatidosis, taeniasis, toxocariasis, etc. have a cross-reaction with fascioliasis. Especially in sheep and cattle.

RESPONSE: we appreciate this comment. During the cattle screening process on farms, coprological examination was used to screen the fecal samples for the presence of parasites eggs. The results of the coprological examination revealed the presence of *F. hepatica* eggs and other parasites, with some samples demonstrating concomitant infections. For the analysis presented in the result section, conducted in order to determine the cutoff points and diagnostic performance measures (sensitivity, specificity, positive and negative predictive values) of the serological tests (Tables 3 and 4 and Figure 2), samples positive for other parasites than *F. hepatica* were excluded (n=67).

Subsequently, these samples were used to explore their absorbance values in relation to the cutoff points established for the serological tests. Furthermore, sensitivity analysis was carried out by adjusting models that considered all samples (those only diagnosed with other parasites were classified in the negative group and those also diagnosed with *F. hepatica* in the positive group), with the aim of evaluating possible changes in the tests performance measures (Supplementary Tables 1 and 2 and Supplementary Figure 1). The description of the positive samples (for fasciolosis and/or other parasites) is now described in detail in lines 292-299.

The reviewer raised a pertinent issue concerning the assessment of cross-infection with other parasites. The utilization of serum samples from animals harboring other parasites is deemed ideal for evaluating cross-infection. Our group is just starting research into diagnostic tools for cattle fasciolosis, and we don't have serum samples from cattle that are positive for other parasites. In addition, COVID-19 has impacted the development of our research. The expectation within our laboratory is to obtain bovine

serum samples demonstrating positivity for other parasites and incorporate them into the sample panel for the development of a lateral flow assays.

8. Table 1. Add OD.

RESPONSE: thank you for mention this. We added the OD term in Table 1 as suggested by the Reviewer.

9. Table 2. Add OD.

RESPONSE: thank you for mention this. We added the OD term in Table 2, in the line correspondent to the cutoff value of each evaluated serodiagnosis test. The other information are diagnostic performance measures.

10. Table 3. Add OD.

RESPONSE: thank you for mention this. We added the OD term in Table 1 as suggested by the Reviewer.

11. Table 4. Add OD.

RESPONSE: thank you for mention this. We added the OD term in Table 4, in the line correspondent to the cutoff value of each evaluated serodiagnosis test. The other information are diagnostic performance measures.

General comment.

The following paragraph should be rewritten. The results show whether it is in the stool or in the serum. Please rewritten clearly. It is difficult to distinguish the eggs of *Fasciola* species by microscopic observation. How was the diagnosis made?

“We found 405 negative and 95 positive results for eggs in the fecal samples. Of 132 the 95 positive animals, 38 (7.6%) were positive for *F. hepatica*, 28 for *F. hepatica* eggs only, and 10 for *F. hepatica* and other parasites: seven also contained eggs of strongylid genera, two *Eimeria* eggs, and one *Strongylidae* and *Eimeria* eggs. The examination also showed that 44 (8.8%) cattle were positive for strongylid genera only and 13 (2.6%) for both strongylid genera and *Eimeria*. Animals positive for other parasites than *F. hepatica* were excluded from the diagnostic performance evaluation of native and recombinant antigens. A total of 139 serum samples were collected from a cattle abattoir located in southern Santa Catarina. The presence of cattle fasciolosis was determined through liver inspection. According to this approach, 10 (7.2%) cattle were diagnosed with *F. hepatica*, with no other parasites investigated during the veterinary inspection.”

RESPONSE: we appreciate this comment and apologize for the for the lack of clarity in the description of the samples used in this research. We rewritten this paragraph, as following: “The coprological examination resulted in 405/500 (81%) negative and 95/500 (19%) positive results. Of the 95 positive results, 28/500 (5.6%) were positive only for *F. hepatica* eggs, and 10/500 (2%) for *F. hepatica* and other parasites: 7/500 (1.4%) also contained *Strongylidae* eggs, 2/500 (0.4%) *Eimeria* eggs, and 1/500 (0.2%) *Strongylidae* and *Eimeria* eggs. The examination also showed that 44/500 (8.8%) cattle were positive only for *Strongylidae* eggs and 13/500 (2.6%) for both *Strongylidae* and

Eimeria eggs. Animals positive for other parasites than *F. hepatica* (n = 67) were excluded from the diagnostic performance evaluation of native and recombinant antigens described below” (lines 292-299 in the results section).

The detection of *F. hepatica* eggs in fecal samples was conducted by proficient laboratory technicians employing the sedimentation protocol. Each sample underwent diagnostic assessment in triplicate, as outlined in lines 136-143 of the methods section.

The section of the manuscript titled "Characteristics of the cattle included in the study" has been revised and restructured. Subtitles have been added to elucidate the number of samples and cattle groups we studied as well as the methodology employed to conduct the comparison with gold standard methods for the diagnostic performance estimation of the ELISA tests.

Re: Spectrum00095-24R1 (Diagnosis of fasciolosis antibodies in Brazilian cattle through enzyme-linked immunosorbent assay (ELISA) employing both native and recombinant antigens)

Dear Dr. GUILHERME DRESCHER:

Your manuscript has been accepted, and I am forwarding it to the ASM production staff for publication. Your paper will first be checked to make sure all elements meet the technical requirements. ASM staff will contact you if anything needs to be revised before copyediting and production can begin. Otherwise, you will be notified when your proofs are ready to be viewed.

Sincerely,
Artem Rogovskyy
Editor
Microbiology Spectrum